# Decentralized federated learning through proxy model sharing

Shivam Kalra [1,2,3], Junfeng Wen[4], Jesse C. Cresswell [1], Maksims Volkovs[1] & H. R. Tizhoosh [2,3,5]

Institutions in highly regulated domains such as finance and healthcare often have restrictive rules around data sharing. Federated learning is a distributed learning framework that enables multi-institutional collaborations on decentralized data with improved protection for each collaborator's data privacy. In this paper, we propose a communication-efficient scheme for decentralized federated learning called ProxyFL, or proxy-based federated learning. Each participant in ProxyFL maintains two models, a private model, and a publicly shared proxy model designed to protect the participant's privacy. Proxy models allow efficient information exchange among participants without the need of a centralized server. The proposed method eliminates a significant limitation of canonical federated learning by allowing model heterogeneity; each participant can have a private model with any architecture. Furthermore, our protocol for communication by proxy leads to stronger privacy guarantees using differential privacy analysis. Experiments on popular image datasets, and a cancer diagnostic problem using high-quality gigapixel histology whole slide images, show that ProxyFL can outperform existing alternatives with much less communication overhead and stronger privacy.

Access to large-scale datasets is a primary driver of advancement in machine learning, with well-known datasets such as ImageNet[1] in computer vision, or SQuAD[2] in natural language processing leading to remarkable achievements. Other domains such as healthcare and finance face restrictions on sharing data, due to regulations and privacy concerns. It is impossible for institutions in these domains to pool and disseminate their data, which limits the progress of research and model development. The ability to share information between institutions while respecting the data privacy of individuals would lead to more robust and accurate models.

In the healthcare domain, for example, histopathology has seen widespread adoption of digitization, offering unique opportunities to increase objectivity and accuracy of diagnostic interpretations through machine learning[3]. Digital images of tissue specimens exhibit significant heterogeneity from the preparation, fixation, and staining protocols used at the preparation site, among other factors.

Without careful regularization, deep models may excessively focus on imaging artifacts and hence fail to generalize on data collected from new sources[4]. Additionally, the need to serve a diverse population including minority or rare groups[5], and mitigate bias[6], requires diverse and multi-centric datasets for model training. Because of specializations at institutions and variability across local populations the integration of medical data across multiple institutions is essential.

However, centralization of medical data faces regulatory obstacles, as well as workflow and technical challenges including managing and distributing the data. The latter is particularly relevant for digital pathology since each histopathology image is generally a gigapixel file, often one or more gigabytes in size. Distributed machine learning on decentralized data could be a solution to overcome these challenges, and promote the adoption of machine learning in healthcare and similar highly regulated domains.

[1]Layer 6 AI, Toronto, ON, Canada. [2]Kimia Lab, University of Waterloo, Toronto, ON, Canada. [3]Vector Institute, Toronto, ON, Canada. [4]Carleton University, School of Computer Science, Ottawa, ON, Canada. [5]Rhazes Lab, Dept. of AI & Informatics, Mayo Clinic, Rochester, MN, USA. ✉e-mail: maks@layer6.ai; tizhoosh.hamid@mayo.edu

Federated learning (FL) is a distributed learning framework that was designed to train a model on data that could not be centralized[7]. It trains a model in a distributed manner directly on client devices where data is generated, and gradient updates are communicated back to the centralized server for aggregation. However, the centralized FL setting is not suited to the multi-institutional collaboration problem, as it involves a centralized third party that controls a single model. Considering a collaboration between hospitals, creating one central model may be undesirable. Each hospital may seek autonomy over its own model for regulatory compliance and tailoring to its own specialty. As a result, decentralized FL frameworks[8] are preferred under such settings.

While it is often claimed that FL provides improved privacy since raw data never leaves the client's device, it does not provide the guarantee of security that regulated institutions require. FL involves each client sending unaudited gradient updates to the central server, which is problematic since deep neural networks are capable of memorizing individual training examples, which may completely breach the client's privacy[9].

In contrast, meaningful and quantitative guarantees of privacy are provided by the differential privacy (DP) framework[10]. In DP, access to a database is only permitted through randomized queries in a way that obscures the presence of individual datapoints. More formally, let $\mathcal{D}$ represent a set of datapoints, and $M$ a probabilistic function, or mechanism, acting on databases. We say that the mechanism is $(\epsilon, \delta)$-differentially private if for all subsets of possible outputs $S \subset$ Range $(M)$, and for all pairs of databases $\mathcal{D}$ and $\mathcal{D}'$ that differ by one element,

$$\Pr[M(\mathcal{D}) \in \mathcal{S}] \le \exp(\epsilon) \, \Pr[M(\mathcal{D}') \in \mathcal{S}] + \delta. \qquad (1)$$

The spirit of this definition is that when one individual's data is added or removed from the database, the outcomes of a private mechanism should be largely unchanged in distribution. This will hold when $\epsilon$ and $\delta$ are small positive numbers. In this case, an adversary would not be able to learn about the individual's data by observing the mechanism's output, hence, privacy is preserved. DP mechanisms satisfy several useful properties, including strong guarantees of privacy under composition, and post-processing[11,12]. However, if an individual contributes several datapoints to a dataset, then their privacy guarantees may be weaker than expected due to correlations between their datapoints. Still, group differential privacy[11] shows that privacy guarantees degrade in a controlled manner as the number of datapoints contributed increases. These properties make DP a suitable solution for ensuring data privacy in a collaborative FL setting.

Unlike centralized FL[7,8] where federated clients coordinate to train a centralized model that can be utilized by everyone as a service, decentralized FL is more suitable for multi-institutional collaborations due to regulatory constraints. The main challenge of decentralized FL is to develop a protocol that allows information passing in a peer-to-peer manner. Gossip protocols[13] can be used for efficient communication and information sharing[14, 15]. There are different forms of information being exchanged in the literature, including model weights[16,17], knowledge representations[18], or model outputs[19, 20]. However, unlike our method, none of these protocols provides a quantitative theoretical guarantee of privacy for participants, and therefore are not suitable for highly regulated domains.

Several other methods have been proposed to achieve decentralization with varying secondary objectives. In Cyclical Weight Transfer (CWT)[21] each client trains a model on local data, then passes that model to the next client in a cyclical fashion. Like standard FL, CWT avoids the need to centralize data and can achieve good performance when strict privacy (à la DP) is not a concern. Split learning[22] enables multiple parties to jointly train a single model with a server such that no party controls the entire model. In our context, the

additional reliance on a central party for inference is undesirable. Finally, swarm learning[23] applies blockchain technology to promote a decentralized, secure network for collaborative training, with one client voted in to act as a central authority each round. Swarm learning does not change the core learning algorithm of FL[7] so it inherits relatively poor model performance when strict privacy measures are applied, and requires homogeneous model architectures.

Each client in our ProxyFL has two models that serve different purposes. They are trained using a DP variant of deep mutual learning (DML)[24] which is an approach for mutual knowledge transfer. DML compares favorably to knowledge distillation between a pre-trained teacher and a typically smaller student[25] since it allows training both models simultaneously from scratch, and provides beneficial information to both models. Federated Mutual Learning (FML)[26] introduces a meme model that resembles our proxy model, which is also trained mutually with each client's private model, but is aggregated at a central server. However, FML is not well-suited to the multi-institutional collaboration setting as it is centralized and provides no privacy guarantee to clients.

Although raw data never leaves client devices, FL is still susceptible to breaches of privacy[27, 28]. DP has been combined with FL to train centralized models with a guarantee of privacy for all clients that participate[29]. By ensuring that gradient updates are not overly reliant on the information in any single training example, gradients can be aggregated centrally with a DP guarantee[30]. We take inspiration from these ideas for ProxyFL.

The main application domain considered in this work is computational pathology. Various articles have emphasized the need for privacy-preserving FL when facing large-scale computational pathology workloads. Li et al.[31] and Ke et al.[32] used FL for medical image augmentation and segmentation. Their method used a centralized server to aggregate selective weight updates that were treated in a DP framework, but they did not account for the total privacy budget expended over the training procedure. Li et al.[33] and Lu et al.[34] built medical image classification models with FL, and added noise to model weights for privacy. However, model weights have unbounded sensitivity, so no meaningful DP guarantee is achieved with these techniques.

In this work, we propose proxy-based federated learning, or ProxyFL (Fig. 1), for decentralized collaboration between institutions which enables training of high-performance and robust models, without sacrificing data privacy or communication efficiency. Our contributions are: (i) a method for decentralized FL in multi-institutional collaborations that is adapted to heterogeneous data sources, and preserves model autonomy for each participant; (ii) incorporation of DP for rigorous privacy guarantees; (iii) analysis and improvement of the communication overhead required to collaborate.

## Results and discussions

In this section, we demonstrate the effectiveness and efficiency of ProxyFL on popular image classification datasets as well as a cancer diagnostic problem. In our experiments, we use the exponential communication protocol of Assran et al.[35], as illustrated in Fig. 2. The communication graph $P^{(t)}$ is a permutation matrix so that each client only receives and sends one proxy per communication round. Each client communicates with its peer that is $2^0, 2^1, \cdots, 2^{\lfloor \log_2(|\mathcal{K}|-1)\rfloor}$ steps away periodically. Sending only one proxy per communication round enables ProxyFL to scale to large numbers of clients. The communication and synchronization is handled via the OpenMPI library[36]. Our code is available at https://github.com/layer6ai-labs/ProxyFL.

### Benchmark image classification
**Datasets and settings.** We conducted experiments on MNIST[37], Fashion-MNIST (FaMNIST)[38], and CIFAR-10[39]. MNIST and FaMNIST have 60k training images of size $28 \times 28$, while CIFAR-10 has 50k RGB

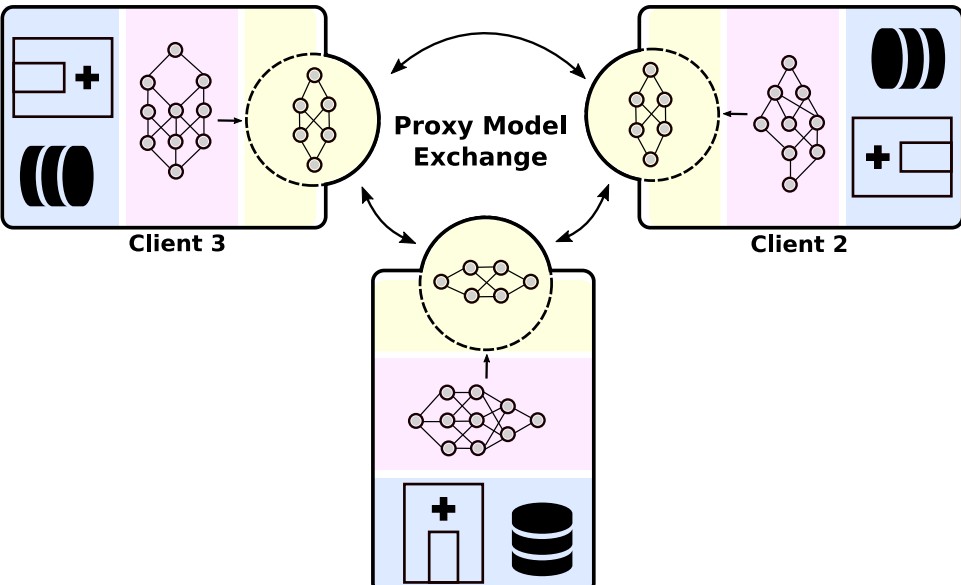

**Fig. 1 | Overall view of the proposed ProxyFL.** ProxyFL is a communication-efficient, decentralized federated learning method where each client (e.g., hospital) maintains a private model, a proxy model, and private data. During distributed training, the client communicates with others only by exchanging their proxy model which enables data and model autonomy. After training, a client's private model can be used for inference.

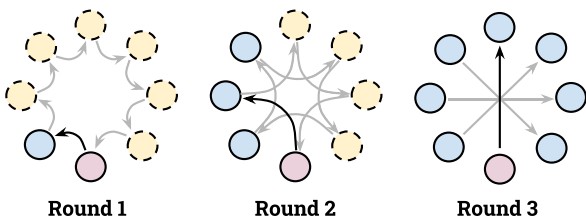

**Fig. 2 | An illustration of the directed exponential graphs used in our experiments.** Dark arrows indicate the communication path of the bottom node at each round, while light arrows show the communication path of others. Solid nodes indicate clients who have received information from the bottom client. After only $\lceil \log_2(|\mathcal{K}|) \rceil$ rounds, all nodes have access to that information.

training images of size $32 \times 32$. Each dataset has 10k test images, which are used to evaluate the model performance. Experiments were conducted on a server with 8 V100 GPUs, which correspond to 8 clients. In each run, every client had 1k (MNIST and FaMNIST) or 3k (CIFAR-10) non-overlapping private images sampled from the training set, meaning only a subset of the available training data was used overall which increases the difficulty of the classification task. To test robustness on non-IID data, clients were given a skewed private data distribution. For each client, a randomly chosen class was assigned and a fraction $p_{major}$ (0.8 for MNIST and FaMNIST; 0.3 for CIFAR-10) of that client's private data was drawn from that class. The remaining data was randomly drawn from all other classes in an IID manner. Hence, clients must learn from collaborators to generalize well on the IID test set.

**Baselines.** We compare our method to FedAvg[7], AvgPush, CWT[21], FML[26], Regular, and Joint training. FedAvg and FML are centralized schemes that average models with identical structure. FML is similar to ProxyFL in that every client has two models, except FML does centralized averaging and originally did not incorporate DP training. AvgPush is a decentralized version of FedAvg using PushSum for aggregation. CWT is similar to AvgPush, but uses cyclical model passing instead of aggregation. In line with prior work[40], we also compare federated schemes with the Regular and Joint settings. Regular training uses the local private datasets without any collaboration. Joint training

mimics a scenario without constraints on data centralization by combining data from all clients and training a single model. Regular, Joint, FedAvg, AvgPush, and CWT use DP-SGD for training their models, while ProxyFL and FML use it for their proxies.

**Implementation details.** Following Shen et al.[26], the private/proxy models are LeNet5/MLP for MNIST and FaMNIST, and CNN2/CNN1 for CIFAR-10. All methods use the Adam optimizer[41] with learning rate of 0.001, weight decay of 1e-4, mini-batch size of 250, clipping $C = 1.0$, and noise level $\sigma = 1.0$. Each round of local training takes a number of gradient steps equivalent to one epoch over the private data. For proper DP accounting, mini-batches are sampled from the training set independently with replacement by including each training example with a fixed probability[42]. The DML parameters $\alpha, \beta$ are set at 0.5 for FML and ProxyFL. We report means and standard deviations based on 5 random seeds. Additional details and results can be found in Sections A and B of the Supplementary Information.

**Results and discussions.** Figure 3 shows the performance on the test datasets. There are a few observations: (i) The private models of ProxyFL achieve the best overall performance on all datasets, even better than the centralized counterpart FML. The improvements of ProxyFL-private over other methods are statistically significant for every dataset ($p$ value < 1e-5). Note that the Joint method serves as an upper bound of the problem when private datasets are combined. (ii) As a decentralized scheme, ProxyFL has a much lower communication cost compared to FML, as shown in Fig. 4. The exponential protocol has a constant time complexity per round regardless of the number of clients, which makes ProxyFL much more scalable. (iii) Decentralized schemes seem to be more robust to DP training, as AvgPush outperforms FedAvg and ProxyFL outperforms FML consistently.

**Ablation**

We ablated ProxyFL to see how different factors affect its performance on the MNIST dataset. Unless specified otherwise, all models, including the private ones in FML and ProxyFL, have the same MLP structure. Additional ablation results can be found in Section B of the Supplementary Information.

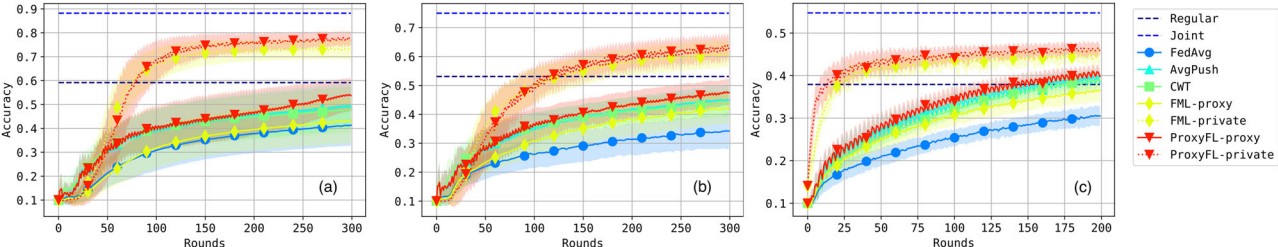

**Fig. 3 | Test performance with differential privacy (DP) training.** Datasets are **a** MNIST, **b** Fashion-MNIST, and **c** CIFAR-10. Each figure reports mean and standard deviation over eight clients for each of five independent runs. Source data are provided as a Source Data file.

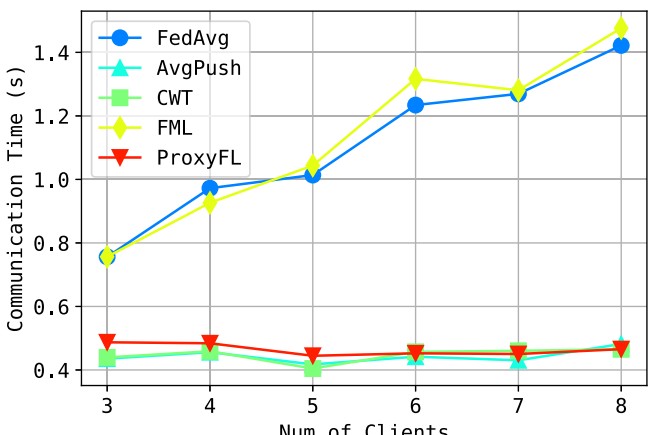

**Fig. 4 | Communication time required as number of clients increases.** Source data are provided as a Source Data file.

**IID versus non-IID**. Figure 5 (Left) shows the performance of the methods with different levels of non-IID dataset skew. $p_{major} = 0.1$ is the IID setting because there are 10 classes. We can see that as the setting deviates from IID, most methods degrade except for Joint training since it unifies the datasets. The private model of ProxyFL is the most robust to the degree of non-IID dataset skew. Note that the proxy model of ProxyFL achieves similar performance to the private model of FML, which is trained without DP guarantees. This indicates that ProxyFL is a scheme that is robust to distribution shifts among the clients.

**Different private architectures**. One important aspect of ProxyFL is that the private models can be heterogeneous—customized to meet the special needs of the individual clients. On the MNIST task we use all four model architectures, one for every two clients (CNN1 and CNN2 are slightly adapted to fit MNIST images). Results are shown in Fig. 5 (Center). We see that different models can achieve very diverse and sub-optimal performances with individual Regular training, while ProxyFL can improve all architectures' performance. The improvements for weaker models are more significant than for stronger models.

**DP versus non-DP**. Next we investigate the effect of DP-SGD on the algorithms. Figure 5 (Right) shows the test accuracies of the different training methods with and without DP-SGD's gradient clipping and noise addition. Clearly, all methods can outperform Regular training when there is no privacy constraint. However, with DP-SGD, centralized methods like FedAvg and FML-proxy perform poorly, even worse than Regular training. ProxyFL-private shows the smallest decrease in performance when DP-SGD is included and remains closest to the upper bound of Joint training.

## Gastrointestinal disease detection

**Dataset and setup**. We applied our method on the Kvasir dataset[43], which is a multi-class image dataset for gastrointestinal disease detection. It consists of 8000 endoscopic images from eight classes such as anatomical landmarks or pathological findings. Each class has 1000 images and each image is resized to 100 × 80 pixels. Following Yang et al.[44], the dataset is partitioned into 6000 training and 2000 test images in each run, and the training set is further distributed into 8 clients by sampling from a Dirichlet distribution with concentration 0.5[45]. The proxy and private models are both the VGG model from Yang et al.[44] with minor adjustments for the resized images. The parameter settings are the same as in "Benchmark image classification" except that the batch size is now 128.

**Results and discussion**. The results are shown in Fig. 6. We can see that centralized schemes like FedAvg and FML-proxy do not learn much during the process, as opposed to their decentralized counterparts AvgPush and ProxyFL-proxy. This demonstrates the effectiveness of the PushSum scheme in this setting. Additionally, ProxyFL-private consistently outperforms FML-private during training, showing that the ProxyFL model is able to provide better learning signal to the private model compared with FML.

## Histopathology image analysis

In this experiment, we evaluated ProxyFL for classifying the presence of lymph node metastases in a tissue sample. The presence of lymph node metastases is one of the most important factors in breast cancer prognosis. The sentinel lymph node is the most likely lymph node to contain metastasized cancer cells and is excised, histopathologically processed, and examined by the pathologist. This tedious examination process is time-consuming and can lead to small metastases being missed[46].

**Dataset**. We considered a large public archive of whole-slide images (WSIs), namely the Camelyon-17 challenge dataset[46], which has previously been used for federated learning in[47]. The dataset is derived from 1399 annotated whole-slide images of lymph nodes, both with and without metastases. Slides were collected from five different medical centers to cover a broad range of image appearance and staining variations. A total of 209 WSIs contain detailed hand-drawn contours for all metastases. The client data for this study was created from Camelyon-17 by choosing WSIs out of these 209 annotated WSIs from four of the institutions (see sample patches in Fig. 7). We used the provided annotations to extract both normal and tumor-containing patches from WSIs. We extracted 512 × 512 pixel patches from WSIs and assigned each patch a binary label (healthy/tumor-containing) based on the provided annotation. A similar number of patches were sampled from the WSIs originating at each of the four institutions, with balanced labels. We then split patches into training and test sets in a rough 80/20 ratio, ensuring that patches in each set originated from non-overlapping patients. The distribution of patches in each split is described in Table 1.

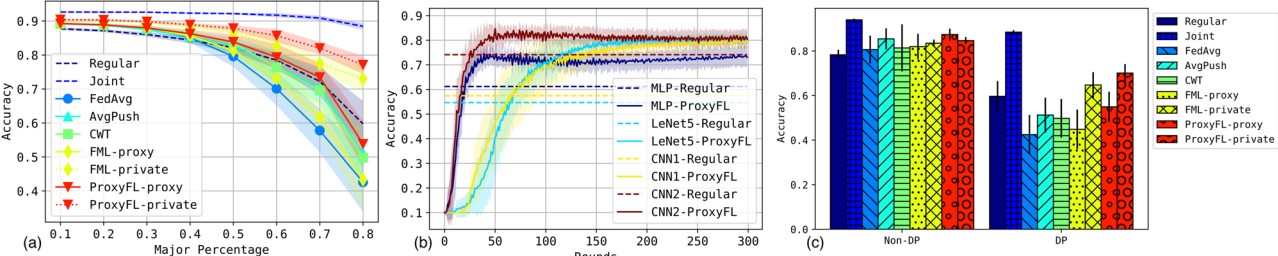

**Fig. 5 | Ablations of our method. a** Accuracy as the non-IID skew of the dataset increases. **b** Accuracy when clients have heterogeneous model architectures, and **c** accuracy with and without differentially private training. Each figure reports mean and standard deviation over eight clients for each of five independent runs. Source data are provided as a Source Data file.

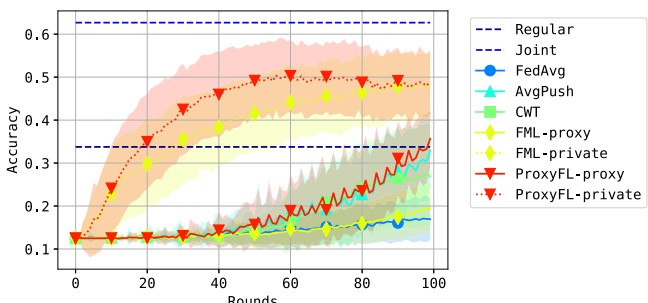

**Fig. 6 | Test accuracy on the Kvasir dataset.** This figure reports mean and standard deviation over eight clients for each of five independent runs. Source data are provided as a Source Data file.

Generalization of models to data collected from diverse sources has become a well-known challenge for applying deep learning to medical applications[48]. The standard method for testing generalization is to evaluate models on external test data which should originate from entirely different institutions than those used for training[49–51]. FL can potentially mitigate generalization problems by increasing the diversity of data that models are exposed to during training. Hence, we merged all four client test sets into a single multi-centric test set used in all model evaluations. From the perspective of a given client, a majority of the test set is external data which requires the trained model to generalize beyond the internal training data in order to show strong performance.

**Models.** For all methods, including both private and proxy models, we used the standard ResNet-18 neural network architecture[52], as implemented in the torchvision package[53], with randomly initialized weights. ResNet architectures use BatchNorm layers[54], which are problematic for the analysis of differential privacy guarantees in DP-SGD training because batch normalization causes each sample's gradient to depend on all datapoints in the batch. As a substitute, we replaced BatchNorm layers with GroupNorm layers[55], and did so across all models for consistency. Finally, we modified the dimensions of the output layer of ResNet-18 to fit our patch size and binary classification requirements.

**Experimental setup.** The experiments were conducted using four V100 GPUs, each representing a client. In each scenario, training was conducted for 30 epochs with a mini-batch size of 32. All methods used the DP settings of $\sigma = 1.4$, $C = 0.7$, and $\delta = 10^{-5}$. Values for $\epsilon$ were then computed per-client based on the training set sizes in Table 1, and are provided in Table 2 (since several datapoints can be associated to the same patient, the formalism of group differential privacy[11] could be applied to account for the correlations between such datapoints. For purposes of demonstration, and because the dataset we use is publicly available, we report $\epsilon$ as if each datapoint came from a distinct patient).

FedAvg, AvgPush, CWT, Regular, Joint, and the proxy models of FML and ProxyFL used a DP-Adam optimizer with learning rate 0.001, whereas the private models of FML and ProxyFL used Adam with the same learning rate. The DML parameters $\alpha$ and $\beta$ were set at 0.3 for both FML and ProxyFL. Finally, for FML and ProxyFL the private models are used to evaluate performance, the central model is used in the case of FedAvg, and for other methods the local models are used.

**Results.** The binary classification results for accuracy and macro-averaged accuracy are reported in Fig. 8 and Table 2. ProxyFL and FML achieve overall higher accuracy throughout training compared to other approaches, due to their private model's ability to focus on local data while extracting useful information about other institutions through proxy models. Notably, FML's performance peaks early and begins to degrade, while ProxyFL continues to improve marginally to the end of training.

## Methods
Our research complies with all relevant ethical regulations; as we have used publicly available data from the Camelyon-17 challenge dataset, no institutional approval was necessary.

ProxyFL, or proxy-based federated learning is our proposed approach for decentralized federated learning. It is designed for multi-institutional collaborations in highly regulated domains, and as such incorporates quantitative privacy guarantees with efficient communication.

### Problem formulation and overview
We consider the decentralized FL setting involving a set of clients $\mathcal{K}$, each with a local data distribution $\mathcal{D}_k, \forall k \in \mathcal{K}$. Every client maintains a private model $f_{\boldsymbol{\phi}_k} : \mathcal{X} \to \mathcal{Y}$ with parameters $\boldsymbol{\phi}_k$, where $\mathcal{X}, \mathcal{Y}$ are the input/output spaces respectively. In this work, we assume that all private models have the same input/output specifications, but may have different structures (this can be further relaxed by including client-specific input/output adaptation layers). The goal is to train the private models collectively so that each generalizes well on the joint data distribution.

There are three major challenges in this setting: (i) The clients may not want to reveal their private model's structure and parameters to others. Revealing model structure can expose proprietary information, increase the risk of adversarial attacks[56], and can leak private information about the local datasets[57]. (ii) In addition to model heterogeneity, the clients may not want to rely on a third party to manage a shared model, which precludes centralized model averaging schemes. (iii) Information sharing must be efficient, robust, and peer-to-peer. To address the above challenges, we introduce an additional proxy model $h_{\boldsymbol{\theta}_k} : \mathcal{X} \to \mathcal{Y}$ for each client with parameters $\boldsymbol{\theta}_k$. It serves as an interface between the client and the outside world. As part of the communication protocol, all clients agree on a common proxy model architecture for compatibility. Proxy models enable architectural independence

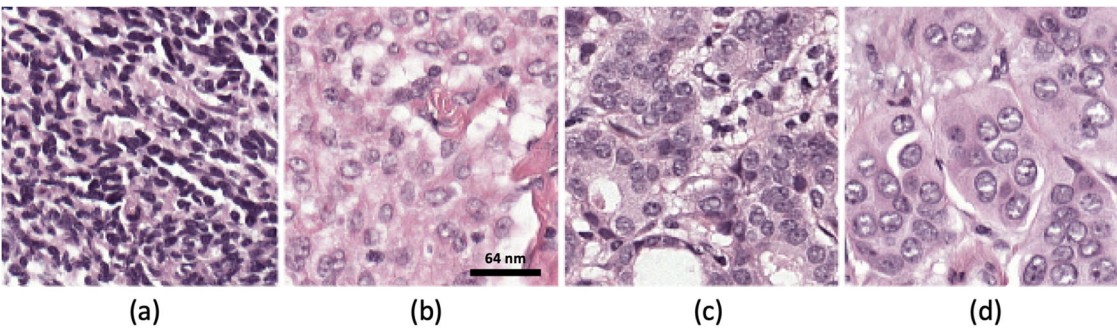

**Fig. 7 | Example histopathology images for model training.** Patches extracted from Camelyon-17 whole-slide images (WSIs): **a, b** healthy tissue, **c, d** from WSIs containing tumors.

among federated clients that may enhance security against white-box adversarial attacks in which the attacker has knowledge of the model architecture[58]. The proxy model is generally smaller than the private model thus reducing communication overhead among clients. Furthermore, diverting all shared information through the proxy allows the private model to be trained without DP. Since DP incurs a loss of utility, the private model, which is ultimately used for inference by the client locally, tends to have much higher accuracy than otherwise.

In every round of ProxyFL, each client trains its private and proxy models jointly so that they can benefit from one another. With differentially private training, the proxy can extract useful information from private data, ready to be shared with other clients without violating privacy constraints. Then, each client sends its proxy to its out-neighbors and receives new proxies from its in-neighbors according to a communication graph, specified by an adjacency matrix $P$ and de-biasing weights $\mathbf{w}$. Finally, each client aggregates the proxies they received, and replaces their current proxy. The overall procedure is shown in Fig. 1 and Algorithm 1. We discuss each step in detail in the subsequent subsections.

## Training objectives

For concreteness, we consider classification tasks. To train the private and proxy models at the start of each round of training, we apply a variant of DML[24]. Specifically, when training the private model for client $k$, in addition to the cross-entropy loss (CE)

$$\mathcal{L}_{\mathrm{CE}}(f_{\boldsymbol{\phi}_k}) := \mathbb{E}_{(\mathbf{x},y) \sim \mathcal{D}_k} \mathrm{CE}[f_{\boldsymbol{\phi}_k}(\mathbf{x}) \| y], \tag{2}$$

DML adds a KL divergence loss (KL)

$$\mathcal{L}_{\mathrm{KL}}(f_{\boldsymbol{\phi}_k}; h_{\boldsymbol{\theta}_k}) := \mathbb{E}_{(\mathbf{x},y) \sim \mathcal{D}_k} \mathrm{KL}[f_{\boldsymbol{\phi}_k}(\mathbf{x}) \| h_{\boldsymbol{\theta}_k}(\mathbf{x})], \tag{3}$$

so that the private model can also learn from the current proxy model. The objective for learning the private model is given by

$$\mathcal{L}_{\boldsymbol{\phi}_k} := (1-\alpha) \cdot \mathcal{L}_{\mathrm{CE}}(f_{\boldsymbol{\phi}_k}) + \alpha \cdot \mathcal{L}_{\mathrm{KL}}(f_{\boldsymbol{\phi}_k}; h_{\boldsymbol{\theta}_k}), \tag{4}$$

where $\alpha \in (0, 1)$ balances between the two losses. The objective for the proxy model is similarly defined as

$$\mathcal{L}_{\boldsymbol{\theta}_k} := (1-\beta) \cdot \mathcal{L}_{\mathrm{CE}}(h_{\boldsymbol{\theta}_k}) + \beta \cdot \mathcal{L}_{\mathrm{KL}}(h_{\boldsymbol{\theta}_k}; f_{\boldsymbol{\phi}_k}). \tag{5}$$

**Algorithm 1. ProxyFL**

---

**Require:** Proxy parameters $\boldsymbol{\theta}_k^{(0)}$, private parameters $\phi_k^{(0)}$, de-biasing weight $w_k^{(0)}$ for client $k$,
　　　　　DML weights $\alpha, \beta \in (0, 1)$, learning rate $\eta > 0$, adjacency matrix $P^{(t)}$

1 **for** *each round* $t = 0, \ldots, T-1$ *at client* $k \in \mathcal{K}$ **do**
2 　　**for** *each local optimization step* **do**
3 　　　　Sample mini-batch $\mathcal{B}_k = \{(\mathbf{x}_i, y_i)\}_{i=1}^B$ from $\mathcal{D}_k$;
4 　　　　Update local proxy and private models:
$$\boldsymbol{\theta}_k^{(t)} \leftarrow \boldsymbol{\theta}_k^{(t)} - \eta \widetilde{\nabla} \widehat{\mathcal{L}}_{\boldsymbol{\theta}_k}(\mathcal{B}_k) \quad \text{\# DP update}$$
$$\phi_k^{(t)} \leftarrow \phi_k^{(t)} - \eta \nabla \widehat{\mathcal{L}}_{\phi_k}(\mathcal{B}_k) \quad \text{\# non-DP update}$$
5 　　**end**
6 　　$\phi_k^{(t+1)} \leftarrow \phi_k^{(t)}$ ;
7 　　Send $\left( P_{k',k}^{(t)} \boldsymbol{\theta}_k^{(t)}, P_{k',k}^{(t)} w_k^{(t)} \right)$ to out-neighbors;
8 　　receive $\left( P_{k,k'}^{(t)} \boldsymbol{\theta}_{k'}^{(t)}, P_{k,k'}^{(t)} w_{k'}^{(t)} \right)$ from in-neighbors;
9 　　Update local proxy $\boldsymbol{\theta}_k^{(t+1)} \leftarrow \sum_{k'} P_{k,k'}^{(t)} \boldsymbol{\theta}_{k'}^{(t)}$;
10 　　Update de-biasing weight $w_k^{(t+1)} \leftarrow \sum_{k'} P_{k,k'}^{(t)} w_{k'}^{(t)}$;
11 　　De-bias $\boldsymbol{\theta}_k^{(t+1)} \leftarrow \boldsymbol{\theta}_k^{(t+1)} / w_k^{(t+1)}$;
12 **end**
13 **return** $\boldsymbol{\theta}_k^{(T)}, \phi_k^{(T)}$;

---

## Table 1 | Distribution of WSI patches across four different institutions

| Training sets | | | | |
| --- | --- | --- | --- | --- |
| | Clients | | | |
| Label | C1 | C2 | C3 | C4 |
| Healthy | 1195 | 1363 | 1727 | 1517 |
| Tumor | 1143 | 1363 | 1210 | 1324 |
| Sum | 2338 | 2726 | 2937 | 2841 |
| Test set | | | | |
| | Clients | | | |
| Label | C1 | C2 | C3 | C4 | Sum |
| Healthy | 322 | 338 | 107 | 344 | 1111 |
| Tumor | 374 | 338 | 624 | 537 | 1873 |
| Sum | 696 | 676 | 731 | 881 | 2984 |

Class labels (healthy vs. tumor-containing) are defined at the WSI level, and are exactly balanced within each Client's overall dataset. Patches in train and test sets are from non-overlapping patients, which forces some imbalance in class labels. The test sets are merged into a single multi-centric dataset for model evaluation.

## Table 2 | (Left) Final performance metrics on the histo-pathology dataset

| Method | Accuracy | Macro-accuracy |
| --- | --- | --- |
| Regular | 0.734 ± 0.105 | 0.703 ± 0.132 |
| AvgPush | 0.776 ± 0.080 | 0.790 ± 0.076 |
| CWT | 0.769 ± 0.087 | 0.783 ± 0.087 |
| FedAvg | 0.786 ± 0.086 | 0.767 ± 0.109 |
| FML | 0.774 ± 0.085 | 0.770 ± 0.010 |
| ProxyFL | 0.808 ± 0.075 | 0.811 ± 0.081 |
| Joint | 0.905 ± 0.030 | 0.913 ± 0.026 |

| Client | Privacy Guarantees $\sigma = 1.4, \delta = 10^{-5}$ |
| --- | --- |
| C1 | $\epsilon = 2.36$ |
| C2 | $\epsilon = 2.17$ |
| C3 | $\epsilon = 2.08$ |
| C4 | $\epsilon = 2.12$ |
| Joint | $\epsilon = 1.00$ |

(Right) Privacy guarantees for each client based on the training set sizes in Table 1. The guarantees are the same across all methods. This table reports mean and standard deviation over 4 clients for each of 15 independent runs. Source data are provided as a Source Data file.

where $\beta \in (0, 1)$. As in DML, we alternate stochastic gradient steps between the private and proxy models.

In our context, mini-batches are sampled from the client's private dataset. Releasing the proxy model to other clients risks revealing that private information. Therefore, each client uses differentially private stochastic gradient descent (DP-SGD)[30] when training the proxy (but not the private model). Let $\mathcal{B}_k = \{(\mathbf{x}_i, y_i)\}_{i=1}^B$ denote a mini-batch sampled from $\mathcal{D}_k$. The stochastic gradient is $\nabla \widehat{\mathcal{L}}_{\boldsymbol{\phi}_k}(\mathcal{B}_k) := \frac{1}{B} \sum_{i=1}^{B} \mathbf{g}_{\boldsymbol{\phi}_k}^{(i)}$ where

$$\mathbf{g}_{\boldsymbol{\phi}_k}^{(i)} := (1 - \alpha) \nabla_{\boldsymbol{\phi}_k} \text{CE}[f_{\boldsymbol{\phi}_k}(\mathbf{x}_i) \| y_i] + \alpha \nabla_{\boldsymbol{\phi}_k} \text{KL}[f_{\boldsymbol{\phi}_k}(\mathbf{x}_i) \| h_{\boldsymbol{\theta}_k}(\mathbf{x}_i)]. \quad (6)$$

$\nabla \widehat{\mathcal{L}}_{\boldsymbol{\theta}_k}(\mathcal{B}_k)$ and $\mathbf{g}_{\boldsymbol{\theta}_k}^{(i)}$ are similarly defined for the proxy. To perform DP training for the proxy, the per-example gradient is clipped, then aggregated over the mini-batch, and finally Gaussian noise is added[30]:

$$\overline{\mathbf{g}}_{\boldsymbol{\theta}_k}^{(i)} := \mathbf{g}_{\boldsymbol{\theta}_k}^{(i)} / \max\left(1, \| \overline{\mathbf{g}}_{\boldsymbol{\theta}_k}^{(i)} \|_2 / C\right),$$
$$\widetilde{\nabla} \widehat{\mathcal{L}}_{\boldsymbol{\theta}_k}(\mathcal{B}_k) := \frac{1}{B} \left( \sum_{i=1}^{B} \overline{\mathbf{g}}_{\boldsymbol{\theta}_k}^{(i)} + \mathcal{N}(0, \sigma^2 C^2 I) \right), \quad (7)$$

where $C > 0$ is the clipping threshold and $\sigma > 0$ is the noise level (see Lines 2–5 in Algorithm 1).

### Privacy guarantee

The proxy model is the only entity that a client reveals, so each client must ensure this sharing does not compromise the privacy of their data. Since arbitrary post-processing on a DP-mechanism does not weaken its $(\epsilon, \delta)$ guarantee[11], it is safe to release the proxy as long as it was trained via a DP-mechanism. DP-SGD as defined in Eq. (7) is based on the Gaussian mechanism[59] which meets the requirement of Eq. (1) by adding Gaussian noise to the outputs of a function $f$ with bounded sensitivity $C$ in $L_2$ norm,

$$M(x) = f(x) + \mathcal{N}(0, \sigma^2 C^2 I). \quad (8)$$

DP-SGD simply takes $f(x)$ to be the stochastic gradient update, with clipping to ensure bounded sensitivity.

Every application of the DP-SGD step incurs a privacy cost related to the clipping threshold $C$, and noise level $\sigma$. A strong bound on the total privacy cost over many applications of DP-SGD is obtained by using the framework of Rényi differential privacy[60,61] to track privacy under compositions of DP-SGD, then convert the result to the language of $(\epsilon, \delta)$-DP as in Eq. (1)[62].

Finally, privacy guarantees are tracked on a per-client basis. In a multi-institutional collaboration, every client has an obligation to

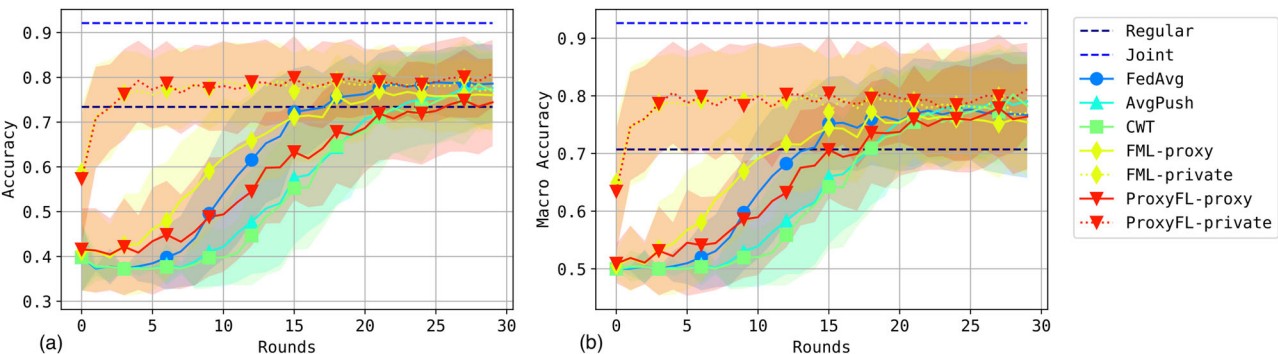

**Fig. 8 | Performance of methods on the histopathology dataset involving four institutions.** For accuracy (**a**) and macro-averaged accuracy (**b**) the mean and standard deviation of performance on the test set is recorded at the end of each epoch. Each figure reports mean and standard deviation over 4 clients for each of 15 independent runs. Source data are provided as a Source Data file.

protect the privacy of the data it has collected. Hence, each client individually tracks the parameters $(\epsilon, \delta)$ for its own proxy model training, and can drop out of the protocol when its prespecified privacy budget is reached. Throughout the paper, we specify $\delta$ based on the dataset size, and compute $\epsilon$.

### Communication efficiency and robustness

The proxies serve as interfaces for information transfer and must be locally aggregated in a way that facilitates efficient learning among clients. One may use a central parameter server to compute the average of the proxies, similar to Shen et al.[26]. However, this will incur a communication cost that grows linearly in the number of clients, and is not decentralized. We propose to apply the PushSum scheme[13, 15] to exchange proxies among clients that significantly reduces the communication overhead.

Let $\Theta^{(t)} \in \mathbb{R}^{|\mathcal{K}| \times d_\theta}$ represent the stacked proxies at round $t$, where the rows are the proxy parameters $\boldsymbol{\theta}_k^{(t)}, \forall k \in \mathcal{K}$. We use $P^{(t)} \in \mathbb{R}^{|\mathcal{K}| \times |\mathcal{K}|}$ to denote the weighted adjacency matrix representing the graph topology at round $t$, where $P_{k,k'}^{(t)} \neq 0$ indicates that client $k$ receives the proxy from client $k'$. Note that $P^{(t)}$ needs to be column-stochastic, but need not be symmetric (bidirectional communication) nor time-invariant (across rounds). Such a $P^{(t)}$ will ensure efficient communication when it is sparse. The communication can also handle asymmetrical connections such as different upload/download speeds, and can adapt to clients joining or dropping out since it is time-varying.

With these notations, every round of communication can be concisely written as $\Theta^{(t+1)} = P^{(t)}\Theta^{(t)}$. Under certain mixing conditions[63], it can be shown that $\lim_{T\to\infty} \prod_{t=0}^{T} P^{(t)} = \boldsymbol{\pi}\mathbf{1}^\top$, where $\boldsymbol{\pi}$ is the limiting distribution of the Markov chain and $\mathbf{1}$ is a vector of all ones. Suppose for now that there is no training for the proxies between rounds, i.e., updates to the proxies are due to communication and replacement only. In the limit, $\boldsymbol{\theta}_k$ will converge to $\boldsymbol{\theta}_k^{(\infty)} = \pi_k \sum_{k'\in\mathcal{K}} \boldsymbol{\theta}_{k'}^{(0)}$. To mimic model averaging (i.e., computing $\frac{1}{|\mathcal{K}|}\sum_{k'\in\mathcal{K}} \boldsymbol{\theta}_{k'}^{(0)}$), the bias introduced by $\pi_k$ must be corrected. This can be achieved by having the clients maintain another set of weights $\mathbf{w} \in \mathbb{R}^{|\mathcal{K}|}$ with initial values $\mathbf{w}^{(0)} = \mathbf{1}$. By communicating $\mathbf{w}^{(t+1)} = P^{(t)}\mathbf{w}^{(t)}$, we can see that $\mathbf{w}^{(\infty)} = \boldsymbol{\pi}\mathbf{1}^\top\mathbf{w}^{(0)} = |\mathcal{K}|\boldsymbol{\pi}$. As a result, the de-biased average is given by $\boldsymbol{\theta}_k^{(\infty)}/w_k^{(\infty)} = \frac{1}{|\mathcal{K}|}\sum_{k'\in\mathcal{K}} \boldsymbol{\theta}_{k'}^{(0)}$.

Finally, recall that the proxies are trained locally in each round. Instead of running the communication to convergence for proxy averaging, we alternate between training (Lines 2–5 in Algorithm 1) and communicating (Lines 7–10) proxies, similar to Assran et al.[35].

### Reporting summary

Further information on research design is available in the Nature Portfolio Reporting Summary linked to this article.

## Data availability

All datasets used are publicly available. MNIST[37], Fashion-MNIST[38], and CIFAR-10[39] are commonly used benchmarks for image classification with machine learning. The Kvasir dataset[43] contains endoscopic images. Our histopathology study used some of the 1399 WSIs from Camelyon-17 ([46]). MNIST URL: https://git-disl.github.io/GTDLBench/datasets/mnist_datasets/Fashion-MNIST: https://www.kaggle.com/datasets/zalando-research/fashionmnistCIFAR-10: https://www.cs.toronto.edu/~kriz/cifar.htmlKvasir: https://www.kaggle.com/datasets/meetnagadia/kvasir-datasetCamelyon-17: https://camelyon17.grand-challenge.org/Data/. The data generated in this study are provided in the Source Data file. All data supporting the findings described in this manuscript are available in the article and in the Supplementary Information and from the corresponding author upon request. Source data are provided with this paper.

## Code availability

Python code of the proposed framework has been made available by Layer 6 AI[64], and will be shared on the "Code and Data" section of Kimia Lab's website (URL: https://kimia.uwaterloo.ca/).

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

## Acknowledgements

S.K. and H.R.T. have been supported by The Natural Sciences and Engineering Research Council of Canada (NSERC). S.K. is also supported by a Vector Institute internship.

## Author contributions

S.K. conceptualized the idea of federated learning through proxy model sharing. J.W. enhanced communication efficiency through the PushSum scheme. J.C.C. contributed to the differential privacy analysis of the method. S.K., J.W., J.C.C. have equal contributions in the experimental analysis and paper writing. The contributions of J.W. were made while J.W. was at Layer 6. M.V. and H.R.T. were involved in the discussions of the approach, and provided critical feedback to the paper. H.R.T. discussed initial ideas with S.K., guided the histopathology experiments and external validations with histopathology images.

## Competing interests

Shivam Kalra is an employee of Roche Diagnostics. Junfeng Wen, Jesse C. Cresswell and Maksims Volkovs were or are employees of Layer 6 AI (TD Bank Group). Hamid R. Tizhoosh does not have any competing interests.

## Ethics

All colleagues who contributed to this research have been added as co-authors. The concrete roles and responsibilities emerged during the research. This research is of global interest.
