## [Peer Review File · Nature Communications]

REVIEWER COMMENTS

Reviewer #1 (Remarks to the Author):

The authors introduce a new method for performing differentially private, decentralized machine learning suitable for multi-institutional collaboration, and benchmark it extensively on both well-known computer vision image classification datasets as well as a real-world dataset of gigapixel histopathology images from the TCGA. Overall, the methodology seems well-motivated, technically sound, clearly written and the work presents a timely contribution in the era of ever-growing digital archives of sensitive medical data that if used carefully, can collectively aid the development of robust and highly accurate machine learning models for both research and potentially real-world clinical workflows. However, in my opinion, there are some important details of experimental study and additional investigations of potential value that are missing from the current iteration of the manuscript, which I hope the authors can address together with a few other miscellaneous questions, provided below. I recommend a minor to moderate revision based on the points below as well as the other reviewers' comments before it can be accepted (subject to the editor's discretion).

Most of my concerns/questions surround the experiments on the computational pathology dataset, which is the only dataset in the study that presents real-world clinical value. The authors identify also identify it as the "main application domain" of their work.

1. There are 32 primary diagnoses in the dataset, are all of them included in the study (since the authors used just a subset of the TCGA)? I.e., is this a 32-class classification problem? For clarify, consider providing a complete table of the all the "classes" and how many slides are in each class for each client.
2. Given the class imbalance, is stratified sampling used to ensure that samples from every class are included in both the training and test set of each client? Does every client's dataset cover all of the classes or are some classes only present for a subset of clients? This should be clarified (once again, consider providing a detailed breakdown of the number of cases in each class for each client for both training/test split).
3. Some patients in the TCGA have multiple slides, is care being taken as to not allow slides from the same patient to fall into both the training and test set?
4. If it is indeed the case that some clients do not contain all the classes, I would appreciate in seeing whether private models for clients which do not directly see those classes, can learn from other clients through FL to meaningfully classify those classes (e.g. if client 1's private training data does not contain class A, but client 2 has class A, can the private model of client 1 learn to classify class A with the help of FL?) This can be a valuable insight as certain rare disease samples/complicated diagnoses will likely only be collected at big, major hospitals so it will be good to know if clients that represent smaller institutions can benefit from these cases too through FL.

5. In A.2, it is stated that Figure 10 displays the “the class distribution per client”, which includes only the top 5 anatomic sites. This is in my opinion confusing, since the goal according to the authors is to classify each slide by its “cancer sub-type (primary diagnosis)” and not merely its anatomic site so the anatomic site should not be called its “class”.

6. How many anatomic sites in the dataset exhibit just a singular primary diagnosis? For those sites the model can effectively “cheat” by just associating the class label with the healthy tissue of the underlying anatomic site (instead of the actual tumor morphology), given that the models are weakly-supervised (training is not confined to just tumor regions). Perhaps this issue can be addressed by including a “normal/healthy” category as well.

7. The use of mean accuracy can be misleading for evaluation on this dataset given the severe class imbalance, i.e. it would be questionable to rank a model that simply achieves high accuracy on the majority diagnoses and completely fails on the 10+ minority diagnoses as better than another model that achieves consistent performance across all classes. I would recommend also reporting the macro-averaged accuracy. However, it is not meaningful to assess performance on minority classes with an extremely small number of cases in the test set, so somehow a cutoff should probably be defined (e.g. only consider diagnoses with >10 slides in the test set of each client).

8. Why are joint training and regular training not provided as baselines for the WSI experiments?

9. Why did the authors chose to represent each WSI with a mosaic + use DeepSet for classification, compared to MIL approaches that consider all regions of the tissue (which is much more popular in the literature)? In any case since this is not a well-known approach, instead of merely dropping a reference, I suggest including all relevant technical details regarding how the mosaic is derived for a given WSI.

10. There are currently no attempt to interpret/investigate the trained models in terms of how they make decisions/their modes of failure, which is crucial in the medical domain. Some example studies include: Campanella, et al. “Clinical-grade computational pathology using weakly supervised deep learning on whole slide images.” *Nat Med* (2019); Lu et al. “Data-efficient and weakly supervised computational pathology on whole-slide images.” *Nat Biomed Eng* (2021). While I understand that this might not be the main focus of the authors’ study, I would like to see some attempt in this matter, e.g. in the case of FL algorithms that train a central model, every client ends up with the same model that behaves the same way (i.e. same prediction, same mode of failure if evaluated on the same set of cases). However, in proxyFL, each client ends-up with its own private model, do they exhibit different modes of failure? How much can we expect these differences to be due to the differences underlying the data distribution of each client vs. strategies for communication/proxy exchange.

11. It is interesting that the authors chose the 32-class primary diagnosis across multiple tumour types to investigate their method, which I believe is very rarely done – most computational studies look at classifying tumour subtypes for different anatomic sites independently or use the TCGA data for survival prediction. This makes sense, since the TCGA contains almost exclusively primary tumors, and the anatomic site is known at the time of inference. Based on the anatomic site information alone (either directly provided to the model, or based on the morphology of the healthy tissue), all diagnoses other than the anatomic-site specific tumour subtypes can presumably be ruled out. Can the authors clarify why they chose to use the pan-cancer TCGA dataset to evaluate classification instead of another multi-institutional dataset such as Camelyon17 for lymph node metastasis detection? For instance, it is what is

used by Andreux et al., “Siloed Federated Learning for Multi-Centric Histopathology Datasets”, 2021. It has the added benefit of coming with detailed annotations for a subset of the cases, which the authors can use in an interpretability study.

PS: I do not see a script/instruction to reproduce the experiments on the histopathology data in the authors' github repo at the time of writing this review – while I do not consider it strictly necessary, it is all the more important that the authors include all relevant details to their experiments in the manuscript.

Other questions/comments:

- I am curious as to what why only a subset of the MNIST/CIFAR training sets are used?

- A relevant work published earlier in Nature is: Warnat-Herresthal, S., Schultze, H., Shastri, K.L. et al. Swarm Learning for decentralized and confidential clinical machine learning. Nature 594, 265–270 (2021). <https://doi.org/10.1038/s41586-021-03583-3>. It would be helpful to readers if the authors can briefly discuss similarities/key differences.

Reviewer #2 (Remarks to the Author):

The authors present an interesting approach for developing communication efficient distributed learning. The authors present results on multiple benchmark computer vision datasets as well as a real-world pathology dataset. Main weaknesses are lack of benchmarking with other distributed learning methods and lack of evaluation on more than 1 real-world dataset.

Major Points:

1) Terminology: Since there is no aggregation in the central server, is it appropriate to call your approach Federated Learning? It is more accurate to describe it more generally as distributed learning as it is more akin to cyclical weight transfer or swarm learning.

2) Comparison with other distributed learning methods: The authors compare with FML and FedAvg. However, there are other methods such as split learning, cyclical weight transfer, and swarm learning. Would recommend benchmarking and comparing with those as well. Especially since communication cost is a sell of the proposed method, would like to see comparison of communication cost across all methods.

2) Line 50 - I wouldn't necessary call DP a "meaningful" guarantee. More accurately, it is a theoretical guarantee. In practice, it is difficult to translate privacy budgets into meaningful insight into how much

of a given client's information is leaked. In this case, there is no results shown about reconstruction attacks. A meaningful guarantee would attempt this and empirically look at what kind of data can be reconstructed.

3) What is the advantage of having 2 models (private and proxy) as opposed to having just one model trained with DP? This is unclear to me and could use some clarification.

4) Since you are exchanging proxies between clients, doesn't that increase the communication cost compared to having a central server? Also makes the approach less scalable when you have a large number of clients

5) One major flaw of this paper is that the authors only present one use-case of real world evaluation with the pathology use-case. Unfortunately, evaluation on MNIST, Fashion-MNIST, and CIFAR-10 are useful for initial development and do not extrapolate well to real use cases. I appreciate the effort to divide up the data to be non-IID. In order to show robustness of the technique, I highly recommend the authors evaluate on at least other real-world use case in other domains, such as radiology, dermatology, and ophthalmology. Would greatly support the effectiveness of the author's approach.

Minor Points:

1) Would have liked to see more in the introduction about about why diverse data is needed, particularly in the medical domain since the authors present a pathology use case. For example, in radiology, image acquisition settings can vary from institution to institution and DL models trained on one institution do not necessarily generalize to another. Furthermore, for rare diseases that may be present in certain geographic regions. Similarly, for minorities and the underserved.

Reviewer #3 (Remarks to the Author):

This paper propose a communication-efficient scheme for decentralized federated learning called ProxyFL. Proxy models allow efficient information exchange among participants using the PushSum method without the need of a centralized server. Experiments show that ProxyFL can outperform existing alternatives with much less communication overhead and stronger privacy.

Pros:

- The proposed ProxyFL is communication-efficient.

- The proposed ProxyFL allows model heterogeneity in FL.
- The proposed ProxyFL leads to stronger privacy guarantees.
- The whole paper is generally well written and easy to follow.

Cons:

- My main concern is novelty. As far as I understand and as acknowledged by the authors in Baselines, ProxyFL is essentially similar as FML, the only difference lies in that ProxyFL makes it decentralized and incorporates DP training by using the existing DP-SGD method. These two add-ons make trivial contributions.
- Swarm learning (Swarm Learning for decentralized and confidential clinical machine learning) is also a well-known decentralized learning framework published by nature, no comparison and even no citation appeared in this work.
- It is expected to show a detailed analysis of the speed-up in terms of efficient communication.
- Non-IID setting did not follow the normal practice in ML, such as by using Dirichlet distribution (Bayesian Nonparametric Federated Learning of Neural Networks).
- The setting of external test data is not realistic.

Overall, this paper is not ready for publication in NCOMMS yet.

We would first like to thank all the reviewers for their constructive feedback and time spent reviewing our submission. We were glad to see that the reviewers found the paper interesting, well motivated, and clearly written. Each of the Reviewer's points have been addressed as follows.

Reviewer 1

1. **There are 32 primary diagnoses in the dataset, are all of them included in the study (since the authors used just a subset of the TCGA)? I.e., is this a 32-class classification problem? For clarify, consider providing a complete table of the all the "classes" and how many slides are in each class for each client.**

For the selected four hospitals, there are total of 26 unique primary diagnoses. However, to develop a consistent model across any hospital from the TCGA dataset, we are using 32 classes as the output from the classification model. We have added a Table 2 in Appendix A, Section A.4 showing all cancer sub-types and their frequency in the dataset we used.

2. **Given the class imbalance, is stratified sampling used to ensure that samples from every class are included in both the training and test set of each client? Does every client's dataset cover all of the classes or are some classes only present for a subset of clients? This should be clarified (once again, consider providing a detailed breakdown of the number of cases in each class for each client for both training/test split).**

No, stratified sampling is not used. To create the training and test sets of each client we used random sampling. Our focus during data splitting was to create datasets that would be realistic in a collaborative learning environment. Hence each client only has data that originated at a single institution. Again, we have now included a more detailed breakdown of the class populations by client in Table 2 (Appendix A, Section A.4).

3. **Some patients in the TCGA have multiple slides, is care being taken as to not allow slides from the same patient to fall into both the training and test set?**

In the revised manuscript we have ensured that slides from a single patient do not fall into multiple datasets after splitting. The overall accuracy across all FL methods declined by around 3-4% for the weaker privacy setting ($\sigma = 0.7$). However, the relative performance remains about the same. We have updated Table 1, Figure 9 (a, b, and c) in Section 4.4 Pan-Cancer Analysis to reflect the changes.

4. **If it is indeed the case that some clients do not contain all the classes, I would appreciate in seeing whether private models for clients which do not directly see those classes, can learn from other clients through FL to meaningfully classify those classes (e.g. if client 1's private training data does not contain class A, but client 2 has class A, can the private model of client 1 learn to classify class A with the help of FL?) This can be a valuable insight as certain rare disease samples/complicated diagnoses will likely only be collected at big, major**

hospitals so it will be good to know if clients that represent smaller institutions can benefit from these cases too through FL.

FL can potentially mitigate generalization problems by increasing the diversity of the data that models are exposed to during training. The prior work [*Federated learning and differential privacy for medical image analysis*, Adnan et al., Nature Scientific Reports 2022] has found FL to achieve comparable results compared to centralized (joint) training. The goal of our study was to compare the relative performance of ProxyFL with other existing FL techniques. We found one particular scenario where the ProxyFL model does not fall into the same mode of failure compared to the central FedAvg model. The details of this scenario have been provided in Appendix B, Section B.2.

- 5. In A.2, it is stated that Figure 10 displays the “the class distribution per client”, which includes only the top 5 anatomic sites. This is in my opinion confusing, since the goal according to the authors is to classify each slide by its “cancer sub-type (primary diagnosis)” and not merely its anatomic site so the anatomic site should not be called its “class”.**

We changed the caption of Figure 10, instead of “class” it is now stated as “anatomical sites”. We now include Table 2 (Appendix A, Section A.4) to provide more details about distributions of WSIs across different primary diagnosis for each client. This provides a complete view on the number of examples of each class available to participating clients. Also see the changes at Line 640 - 642.

- 6. How many anatomic sites in the dataset exhibit just a singular primary diagnosis? For those sites the model can effectively “cheat” by just associating the class label with the healthy tissue of the underlying anatomic site (instead of the actual tumor morphology), given that the models are weakly-supervised (training is not confined to just tumor regions). Perhaps this issue can be addressed by including a “normal/healthy” category as well.**

There are multiple tumor types with a single cancer subtype, and these are all displayed in Table 2 (Appendix A.4). Since we are performing a relative comparison of different FL methods on the same datasets, we did not remove WSIs with these tumor types. More changes added at Section 4.4, Line 345 - 348.

- 7. The use of mean accuracy can be misleading for evaluation on this dataset given the severe class imbalance, i.e. it would be questionable to rank a model that simply achieves high accuracy on the majority diagnoses and completely fails on the 10+ minority diagnoses as better than another model that achieves consistent performance across all classes. I would recommend also reporting the macro-averaged accuracy. However, it is not meaningful to assess performance on minority classes with an extremely small number of cases in the test set, so somehow a cutoff should probably be defined (e.g. only consider diagnoses with >10 slides in the test set of each client).**

We have reevaluated our benchmark experiments using macro-averaged accuracy which is more sensitive to performance on minority classes. Results and discussion are

given in Appendix B.1 and Figure 11. In summary, there is very little difference between accuracy and macro-accuracy in our experiments even though client training datasets have 80% of samples from a single class (for MNIST and Fashion-MNIST).

8. Why are joint training and regular training not provided as baselines for the WSI experiments?

Joint training is unrealistic for the WSI experiments, since raw patient data cannot be centrally pooled in a real world application. We chose to only compare methods that could plausibly be used in practice. However, we have expanded the benchmarked methods to include Cyclical Weight Transfer. For comparison to regular training we have added references to existing literature comparing FL to centralized training in clinical applications [*Federated learning and differential privacy for medical image analysis*, Adnan et al., Nature Scientific Reports 2022], see Section 4.4 - Experimental Setup (Line 329 - 330).

9. Why did the authors chose to represent each WSI with a mosaic + use DeepSet for classification, compared to MIL approaches that consider all regions of the tissue (which is much more popular in the literature)? In any case since this is not a well-known approach, instead of merely dropping a reference, I suggest including all relevant technical details regarding how the mosaic is derived for a given WSI.

Given the high resolution nature of gigapixel WSIs, it is common to extract regions of pixels and classify them as a set. In Multiple Instance Learning applied to gigapixel WSIs the extracted patches often cover only portions of the full tissue sample - see for example recent works [*Dual-Stream Multiple Instance Learning Network for Whole Slide Image Classification With Self-Supervised Contrastive Learning*, Li et al., CVPR 2021] and [*TransMIL: Transformer based Correlated Multiple Instance Learning for Whole Slide Image Classification*, Shao et al., NeurIPS 2021]. We used an unsupervised method to extract patches from WSIs, and a DeepSet-based architecture for the MIL component, and have added more detail to the manuscript, particularly in Appendix A.2, about the method instead of only citing the original source.

10. There are currently no attempt to interpret/investigate the trained models in terms of how they make decisions/their modes of failure, which is crucial in the medical domain. Some example studies include: Campanella, et al. "Clinical-grade computational pathology using weakly supervised deep learning on whole slide images." Nat Med (2019); Lu et al. "Data-efficient and weakly supervised computational pathology on whole-slide images." Nat Biomed Eng (2021). While I understand that this might not be the main focus of the authors' study, I would like to see some attempt in this matter, e.g. in the case of FL algorithms that train a central model, every client ends up with the same model that behaves the same way (i.e. same prediction, same mode of failure if evaluated on the same set of cases). However, in proxyFL, each client ends-up with its own private model, do they exhibit different modes of failure? How much can we expect these

differences to be due to the differences underlying the data distribution of each client vs. strategies for communication/proxy exchange.

Investigating a trained model's decision making is important before deploying a model into real-world use, and we acknowledge this as an interesting future direction to explore, but our focus in this paper is on the challenges of collaborative training. In an effort to address the question we have added a section to Appendix B.2 on a particular failure mode that is observed. We found that the central model trained with FedAvg has a mode of failure while distinguishing cancer subtypes for lung slides - i.e. classifying LUAD versus LUSC. The model predicts LUSC for all the given WSIs. We did not observe this mode of failure when ProxyFL was used. Please refer to Appendix B, Section B.2 (Comparison of Mode of Failure) for more details.

11. It is interesting that the authors chose the 32-class primary diagnosis across multiple tumour types to investigate their method, which I believe is very rarely done – most computational studies look at classifying tumour subtypes for different anatomic sites independently or use the TCGA data for survival prediction. This makes sense, since the TCGA contains almost exclusively primary tumors, and the anatomic site is known at the time of inference. Based on the anatomic site information alone (either directly provided to the model, or based on the morphology of the healthy tissue), all diagnoses other than the anatomic-site specific tumour subtypes can presumably be ruled out. Can the authors clarify why they chose to use the pan-cancer TCGA dataset to evaluate classification instead of another multi-institutional dataset such as Camelyon17 for lymph node metastasis detection? For instance, it is what is used by Andreux et al., “Siloed Federated Learning for Multi-Centric Histopathology Datasets”, 2021. It has the added benefit of coming with detailed annotations for a subset of the cases, which the authors can use in an interpretability study.

Large scale, multi-institution datasets are not yet common in digital histopathology, though there are notable examples, namely Camelyon-17 and TCGA. Of those two, TCGA contains more images, and has a wider distribution of classes which presents a greater challenge for collaborative FL approaches - each client's dataset is highly non-IID. In comparison Camelyon-17 contains 1399 WSIs, whereas we used 5616 from TCGA. Both datasets are very valuable resources to the research community, but TCGA fit our needs better.

12. I do not see a script/instruction to reproduce the experiments on the histopathology data in the authors' github repo at the time of writing this review – while I do not consider it strictly necessary, it is all the more important that the authors include all relevant details to their experiments in the manuscript.

We provided scripts for the benchmark experiments because they are self-contained and can be run immediately by the reviewers or readers with few resource requirements. TCGA data must be manually retrieved by the user and requires significant storage and compute resources. In addition to greater detail on the histopathology experiment setup,

we can provide code to reproduce the experiments.

13. I am curious as to what why only a subset of the MNIST/CIFAR training sets are used?

These benchmark datasets are simple enough that basic neural networks can achieve good performance even on a fraction of the training data. We tried to create a reproducible experimental setting where non-collaborative training performed poorly. We used a subset of these datasets to begin with, and further divided them into small client datasets. This also reduces the resource requirements of running the experiments which should allow greater accessibility to researchers and reviewers for reproducing our results. We added our reasoning to Section 4.1 under “Datasets & settings” (Line 230-231).

14. A relevant work published earlier in Nature is: Warnat-Herresthal, S., Schultze, H., Shastry, K.L. et al. Swarm Learning for decentralized and confidential clinical machine learning. Nature 594, 265–270 (2021). <https://doi.org/10.1038/s41586-021-03583-3>. It would be helpful to readers if the authors can briefly discuss similarities/key differences.

Swarm Learning (SL) [*Swarm Learning for decentralized and confidential clinical machine learning*, Warnat-Herresthal et al., Nature 2021] is an approach to decentralized FL that has been proposed for medical applications. SL is based on FedAvg but innovates by enabling a decentralized setting. The core learning algorithm of SL is still FedAvg, but whereas in FedAvg the update averaging is done by a fixed 3rd party server, in SL clients will nominate one participant each round to act as a central server (“leader”). The learning algorithm and need for homogeneous models are unchanged. The innovation of SL is in using blockchain and encryption to manage the nomination of a leader, and secure communication between participants. However this comes with significant overhead; in SL each client is required to run four dedicated servers for ML training, blockchain communication, constant identity verification, and hosting a software license. SL provides network security, but does not give a guarantee of privacy to individuals whose data is used.

In comparison, ProxyFL enables decentralized FL with four key components that change the learning algorithm. First, clients can maintain private models with heterogeneous and secret architectures because all information transfer is done through standardized proxy models. Second, information transfer between clients is achieved through mutual learning, where knowledge is exchanged between a private model and the current proxy. Third, efficient and decentralized communication is managed through the PushSum protocol with exponential communication graphs. Finally, a guarantee of privacy is provided to individuals because the proxy models, being the only information released by clients, are trained with DP-SGD.

We also point out that SL is being marketed as a commercial software and requires the purchase of a license to use. Since it is not freely available to the research community,

we have not run the SL code to compare approaches. However, since SL does not change the learning algorithm underpinning FedAvg, the accuracy of models trained with SL should be identical to FedAvg, which we have benchmarked against. We have added a mention of SL in Section 2 Related Work (Line 85-95).

Reviewer 2

- 1. Terminology: Since there is no aggregation in the central server, is it appropriate to call your approach Federated Learning? It is more accurate to describe to describe it more generally as distributed learning as it is more akin to cyclical weight transfer or swarm learning.**

We see FL as a group of methods designed to train models on data that cannot be centralized. The initial proposed solutions to the FL problem, such as FedAvg, use a server that centralizes information other than the raw data. ProxyFL still addresses the FL problem that raw data cannot be centralized, but does so without any central authority, which we suspect is preferable in many cases.

- 2. Comparison with other distributed learning methods: The authors compare with FML and FedAvg. However, there are other methods such as split learning, cyclical weight transfer, and swarm learning. Would recommend benchmarking and comparing with those as well. Especially since communication cost is a sell of the proposed method, would like to see comparison of communication cost across all methods.**

In the initial submission we benchmarked against several methods that were either very well known (FedAvg), or had similar characteristics to ProxyFL (FML, AvgPush). Please see our response about Swarm Learning above (Reviewer 1, Question 14). Cyclical Weight Transfer (CWT) [*Distributed deep learning networks among institutions for medical imaging*, Chang et al., JAMIA 2018] is another method which is comparable to ours, and we have updated our experiments to include it. CWT is similar to our benchmark AvgPush in that clients directly exchange their model weights. This method will incur higher communication cost to ProxyFL when the private model is larger than the proxy model, see Figure 8 (d) in Section 4.3 of the updated manuscript. ProxyFL outperforms CWT because in ProxyFL DP-SGD does not need to be used on the private model that is eventually deployed. The following sections/figure have been updated to include CWT benchmarking results – Section 4.1 Figures 3, 5, 7, and Section 4.4 Figure 9 a,b,d.

Split Learning is another interesting technique that could have applications for collaborative training of models without sharing raw data. The biggest difference between Split Learning and the FL approaches we have tested is that in Split Learning

no single entity has a full copy of the model being trained. In particular, the clients do not have full control of their copies of the model, and cannot test or deploy it without the central server being online. We expect that regulated institutions would require control over their models to meet oversight requirements. We have added mentions of all these methods in Section 2 Related Work (Line 85-95).

- 3. Line 50 - I wouldn't necessary call DP a "meaningful" guarantee. More accurately, it is a theoretical guarantee. In practice, it is difficult to translate privacy budgets into meaningful insight into how much of a given client's information is leaked. In this case, there is no results shown about reconstruction attacks. A meaningful guarantee would attempt this and empirically look at what kind of data can be reconstructed.**

Differential privacy is theoretical, in that there is an underlying mathematical theory that provides rigorous guarantees. It is also meaningful, in that a differential privacy provides *worst-case* guarantees that cannot be weakened through external attacks. This contrasts with older approaches to privacy such as anonymization, for which joining with external data can potentially deanonymize supposedly private records. We appreciate the practical point of view that differential privacy guarantees could be made more meaningful by showing they increase robustness against attacks. There is an active line of research that tests the susceptibility of models trained with DP-SGD to reconstruction attacks and membership inference attacks. The paper [*Auditing Differentially Private Machine Learning: How Private is Private SGD?*, Jagielski et al., NeurIPS 2020] uses realistic privacy attacks to empirically test how much privacy is afforded by DP-SGD, and very recently [*Adversary instantiation: Lower bounds for differentially private machine learning*, Nasr et al., IEEE SP 2021] demonstrate worst-case attacks that achieve the maximum privacy violation allowed by the theoretical guarantees of DP-SGD but do not breach them. We have added a discussion of these important matters in Section 5 Conclusion and Future Work (Line 373-380).

- 4. What is the advantage of having 2 models (private and proxy) as opposed to having just one model trained with DP? This is unclear to me and could use some clarification.**

There are several advantages in performance, communication efficiency, personalization, and privacy.

In prior FL systems the single model needs to be trained locally then shared with either a central server or peers. This requires all participants to agree on a common model architecture, and clients should train their copies with DP-SGD to provide privacy guarantee. It has been well established that using DP-SGD lowers the utility (viz. accuracy) of models, leading to an underperformance of previous FL systems. ProxyFL uses two models, the private model and proxy model. Because the private model is never shared, it does not need to be trained with DP-SGD to ensure privacy. On benchmarks and real-world settings, we observed the ProxyFL private model outperformed other approaches.

Since the proxy model is only used for information transfer and not eventual deployment, we found it acceptable to use a smaller architecture, which means less data to transfer thereby increasing communication efficiency

Perhaps the most significant advantage comes from personalization which is a qualitatively distinct advantage of ProxyFL. Because proxy models provide the common language for information transfer, clients are free to design private models in any way they see fit. Private model architectures can be completely secret, and personalized to a client's needs and data. We demonstrated that all clients can use different architectures and still benefit from collaboration in Fig. 6.

Finally, and with your previous question in mind, having a private model with secret architecture limits the types of adversarial attacks that clients are exposed to. *White-box* attacks, in which an attacker has information about the model including its architecture, have been shown to be more threatening than *black-box* attacks, where only inputs and outputs can be observed but not the model itself [*Towards Evaluating the Robustness of Neural Networks*, Carlini and Wagner, IEEE SP 2017]. By communicating through proxy models, ProxyFL allows users to keep their private model details a secret, limiting the attack surface.

We have expanded on these points in Section 3.1 (Line 137-143).

5. Since you are exchanging proxies between clients, doesn't that increase the communication cost compared to having a central server? Also makes the approach less scalable when you have a large number of clients

No, peer-to-peer exchange of proxy models is more efficient than communication with a central server, as we demonstrated in Fig. 4, and Fig. 9 (d). In centralized systems the central server can become a bottleneck for communication, having to manage communication with all clients simultaneously. In ProxyFL peer-to-peer communication means no single entity needs to communicate with all other parties at once. We observed that ProxyFL has constant per-client communication times as the number of clients increases, whereas centralized systems have linearly increasing communication times because all information needs to pass through one server.

6. One major flaw of this paper is that the authors only present one use-case of real world evaluation with the pathology use-case. Unfortunately, evaluation on MNIST, Fashion-MNIST, and CIFAR-10 are useful for initial development and do not extrapolate well to real use cases. I appreciate the effort to divide up the data to be non-IID. In order to show robustness of the technique, I highly recommend the authors evaluate on at least other real-world use case in other domains, such as radiology, dermatology, and ophthalmology. Would greatly support the effectiveness of the author's approach.

We appreciate that you value benchmark datasets like MNIST, Fashion-MNIST, and CIFAR-10 for development, and agree that such results do not always translate into real-world settings. We have conducted additional experiments on one more medical datasets - Kvasir - a dataset of endoscopic images labeled by medical doctors. A full discussion of each new experiment is provided in the updated draft (Section 4.3, Fig. 8).

- 7. Would have liked to see more in the introduction about why diverse data is needed, particularly in the medical domain since the authors present a pathology use case. For example, in radiology, image acquisition settings can vary from institution to institution and DL models trained on one institution do not necessarily generalize to another. Furthermore, for rare diseases that may be present in certain geographic regions. Similarly, for minorities and the underserved.**

Yes, this is a great suggestion. Diverse data is needed in the medical domain for many reasons, including the heterogeneity of data collection equipment as we mentioned in Section 1. In addition, there is the tendency of machine learning models to pick up spurious correlations which diverse data can mitigate [*Problems in the deployment of machine-learned models in health care*, Cohen et al., CMAJ 2021], the need to serve a diverse population including minority groups [*Ensuring machine learning for healthcare works for all*, McCoy et al., BMJ Health & Care Informatics 2020], and to mitigate bias [*Mitigating bias in machine learning for medicine*, Volkinger et al., Nature Communications Medicine 2021].

We have added more discussion of this point to Section 1 - Introduction, but also see relevant updates in Section 4.4 - External Testing.

Reviewer 3

- 1. My main concern is novelty. As far as I understand and as acknowledged by the authors in Baselines, ProxyFL is essentially similar as FML, the only difference lies in that ProxyFL makes it decentralized and incorporates DP training by using the existing DP-SGD method. These two add-ons make trivial contributions.**

The main novelty of our paper is a FL framework that is simultaneously decentralized, differentially private, and communication efficient. Although there is prior work that satisfies each of these properties individually, it is non-trivial to ensure them within a single framework. As you have pointed out, the closest previous work, FML, is neither decentralized, nor differentially private, and is less communication efficient than ProxyFL (see Figs. 4, 9d).

We discussed under the heading “Computational Pathology” four previous proposals for using differentially private FL in a medical setting. We have explained why their methods of applying DP are flawed and do not provide a mathematically sound DP guarantee. Clearly then, a correct application of differential privacy is non-trivial, and our work demonstrates to the community one correct way forward, namely DP-SGD.

2. Swarm learning (Swarm Learning for decentralized and confidential clinical machine learning) is also a well-known decentralized learning framework published by nature, no comparison and even no citation appeared in this work. Thank you for pointing out this work. Please see our comment to Reviewer 1, Question 14.

3. It is expected to show a detailed analysis of the speed-up in terms of efficient communication.

As part of our experiments we investigated communication costs for ProxyFL compared to other approaches (Figs. 4, 9d). Decentralized approaches like ProxyFL tend to have a constant per-client communication cost, while centralized approaches including FML have linearly increasing cost with the number of clients. Please also see our discussion under Reviewer 2, Question 5.

4. Non-IID setting did not follow the normal practice in ML, such as by using Dirichlet distribution (Bayesian Nonparametric Federated Learning of Neural Networks).

The Dirichlet distribution is a suitable tool for creating non-IID data splits, but it is far from the only way of doing so. The data splitting technique in [*Bayesian Nonparametric Federated Learning of Neural Networks*, Yurochkin et al., ICML 2019] is relevant, but is not fundamentally different from our splitting method of selecting classes in differing proportions per client. The data split shown in Table 1 is naturally non-IID since it is based on data collected from different medical institutions, and the use of Dirichlet sampling would not be appropriate here because we do not intend to mix the institution's data. However, to demonstrate that both approaches can be useful, we created non-IID datasets for the new Kvasir experiment in Section 4.3 by sampling from a Dirichlet distribution.

5. The setting of external test data is not realistic.

We respectfully, but strongly disagree that external testing is unrealistic. It has become a well known problem in the field of machine learning for medical applications that models trained on data from a single source will often fail to generalize to data collected from another source, see for example the papers [*Methodologic Guide for Evaluating Clinical Performance and Effect of Artificial Intelligence Technology for Medical Diagnosis and Prediction*, Park and Han, Radiology 2016], [*Evaluating reproducibility of AI algorithms in digital pathology with DAPPER*, Bizzego et al. PLOS Comp. Bio. 2019], [*Key challenges for delivering clinical impact with artificial intelligence*, Kelly et al., BMC Medicine 2019], and [*Assessing Radiology Research on Artificial Intelligence: A Brief Guide for Authors, Reviewers, and Readers—From the Radiology Editorial Board*, Bluemke et al., Radiology 2019]. We have expanded our discussion of the external testing approach in Section 4.4 to include these references.

The aim of FL in this setting is to improve generalizability by increasing the diversity of information accessible to each client. External test data provides one perspective on the question of generalizability.

REVIEWER COMMENTS

Reviewer #1 (Remarks to the Author):

I thank the authors for taking their time to include additional experiments, revising the manuscript, and replying to the reviewers' comments. However, I do feel several of my points of confusion were not adequately addressed in the current round of revision.

Overall, my assessment is that the usage of the 5616 slides from the TCGA and doing a "32"-class classification problem is inappropriate for benchmarking for multiple reasons (see my detailed comments below). Based on the authors' response to my point 11, it appears it was chosen simply because the authors wanted to make it seem like they did a large-scale study. When in fact the multi-institutional Camelyon-17 challenge seems to be far more suitable, and was used by a previous study for Federated learning (Andreux et al. 2021). Another sensible option, is that instead of considering the 32 tumor subtypes in TCGA, previous studies have selected just a few specific disease models from the TCGA and did survival analysis (e.g. see "Federated Survival Analysis with Discrete-Time Cox Models") under the framework of federated learning.

Point 1. If I understand correctly, you make each model output predictions for 32-classes even though you only have training/test data for 26-classes? While I agree that this should not affect the relative comparison, this is confusing for readers. It probably should be simply stated as a 26-class problem in the manuscript.

Also, is random sampling used to create train/test sets? I do not think that is the right approach in this case. Given some classes have so few cases (e.g. SARC has less than 15 cases in total), if you randomly sample your training/test splits, it is very likely some of the minority classes will not appear at all in the test set. Can the authors clarify if this is indeed the case? i.e., I do not think it's enough that the authors just provide a count of cases in each class across the entire dataset, instead please specify how many cases are in the training and in the test of each class. This is how it is done in every major published computational pathology study and should be also followed here. After all, there is no reason to include minority classes in your training set/problem if you do not plan on evaluating on them.

Point 7. Given the large class imbalance in the histology dataset (e.g. some classes have < 15 samples in total, and some have nearly 1000), I had specifically asked for the report of macro-average (i.e. balanced accuracy) on the histology dataset. Yet for some reason the authors chose to only report balanced accuracy on the benchmark datasets (figure 12)? When the class-imbalance is this significant, the single accuracy metric is simply not sufficient to characterize the performance of the models. For instance, a model that simply predicts every case as the single predominant class will appear to do very well

compared to the competition when in fact it is useless. Given that this is supposed to be an application to a real-world problem, I would suggest that the authors at least report:

- accuracy for each class
- macro-averaged accuracy over all classes

Another suggestion would be to simply remove any classes with < 10 or < 5 cases in the test set from the problem, since we cannot reliably assess the model's performance on those classes.

Point 9. I am quite familiar with WSI-classification and both references the authors mentioned (Dual-Stream Multiple Instance Learning Network and TransMIL) do not sample patches (i.e. use a mosaic representation) the way the authors do in this work. Both approaches presumably consider all tissue regions in the slide for the classification task (similarly, the other works I mentioned, i.e. Campanella et al 2019, and Lu et al. 2021 also process/consider the entire tissue region for classification). Therefore I do appreciate that the authors added additional explanation about the method they used to the manuscript.

Point 11. As I alluded to and the authors also acknowledged, there are a number of anatomic sites in the TCGA dataset considered that only have a single associated tumor subtype and therefore the model can just infer the label from normal tissue region. It is also not a useful problem to solve since the anatomic site is presumably given. Why not just remove these classes/slides from the problem? This is probably the main reason that every computational study for classification I have seen restrict their attention to classifying between multiple subtypes that may be present at a given anatomic subtype. Since you already used multiple benchmark datasets, and want to use at least one real-world problem that is clinically relevant, then I think it would be much more appealing to readers with clinical or computational pathology background that the problem the authors proposed to solve is actually a meaningful one and is suitably framed.

Lastly judging from figure 9 b), the performance of ProxyFL vs. other approaches appear to be the similar under strong differential privacy guarantee, and ProxyFL actually does statistically significantly worse than CWT under weak privacy? Can authors confirm if this is indeed the case? So is the only advantage of ProxyFL faster communication then? However, looking at 9 d), it is not like ProxyFL is an order of magnitude faster either, so perhaps the authors want to summarize and make a stronger case by clarifying what are the clear advantages of the proposed method over the competing approaches.

Reviewer #4 (Remarks to the Author):

The authors have addressed most of the previous comments satisfactorily, with some remaining points included below.

1. With regard to Rev#2 Q#1 (terminology), the fact that the authors consider FL as a group of methods should be clearly defined in the manuscript, as otherwise might mislead the reader.

2. In response to Rev#2 Q#2 the authors point to additional text that they provide in Section 2 “Related Work (Line 85-95)”: “However, unlike our method, none of these protocols provides a guarantee of privacy for participants, and therefore cannot be used safely in highly regulated domains”. It is unclear what the authors mean by “safely”. Please either clarify, or reword.

3. In response to Rev#2 Q#3 the authors point to additional text that they provide in “Conclusion and Future Work (Line 373-380)”: “Finally, while differential privacy provides theoretical guarantees on the protection of privacy based on mathematical analysis, it is important to validate the effectiveness of these guarantees in practice. To be meaningful, such guarantees should demonstrably reduce the susceptibility of systems to reconstruction and membership inference attacks [Bhowmick et al., 2018]. Recent works use realistic privacy attacks to empirically test how much privacy is afforded by DP-SGD [Jagielski et al., 2020], while Nasr et al. [2021] demonstrate worst-case attacks that achieve the maximum privacy violation allowed by the theoretical guarantees of DP-SGD, but do not breach them. Testing and validation of privacy guarantees should be incorporated into model development and deployment processes.”

I would also mention here that DP guarantees in the paper use accounting based on to what degree a single data sample can affect the overall outcome. These guarantees do not speak to vulnerability of attacks that make use of correlations across the data samples. A simple example of this is the fact that $\epsilon=0$ DP (‘perfect DP’) will leak a lot of information about my single data sample if it is applied to a dataset that consists solely of multiple copies of that single sample (since the DP analysis only compares query outcomes with and without a single sample removed). Generally, for this reason, it is more possible than the DP guarantee may suggest extracting information from the result of a DP algorithm (as defined in the paper) when there are strong correlations regarding this information across samples. This is another way in which ϵ , δ numbers may not provide the meaning of privacy that one may expect, and the reader should be aware of this.

4. Finally, in addition to the original review comments, I do not see the authors explaining how the adjacency matrix (P_t in Algorithm 1) is selected for the various experiments. Whether it was randomly initialized (i.e. how P_0 is chosen), constant over time (i.e. $P_t=P_0$ for all t), and/or evolved in some specified way.

Reviewer comments are highlighted.
Our responses are in plaintext.
The revised sections in the manuscript are highlighted in RED.

Review #1

I thank the authors for taking their time to include additional experiments, revising the manuscript, and replying to the reviewers' comments. However, I do feel several of my points of confusion were not adequately addressed in the current round of revision.

Overall, my assessment is that the usage of the 5616 slides from the TCGA and doing a "32"-class classification problem is inappropriate for benchmarking for multiple reasons (see my detailed comments below). Based on the authors' response to my point 11, it appears it was chosen simply because the authors wanted to make it seem like they did a large-scale study. When in fact the multi-institutional Camelyon-17 challenge seems to be far more suitable, and was used by a previous study for Federated learning (Andreux et al. 2021). Another sensible option, is that instead of considering the 32 tumor subtypes in TCGA, previous studies have selected just a few specific disease models from the TCGA and did survival analysis (e.g. see "Federated Survival Analysis with Discrete-Time Cox Models") under the framework of federated learning.

Response: Thank you for clarifying the concerns around the 32-class classification problem from our previous submissions. Given all of the reviewer's convincing points, we have heeded your/their advice and revised the histopathology experiment to use the Camelyon-17 dataset instead of TCGA. In the new submission, please find that Section 4.4 now demonstrates ProxyFL on a binary classification problem for predicting healthy or tumor-containing histopathology images, in the direction established by [Andreux et al. "Siloed Federated Learning for Multi-Centric Histopathology Datasets", 2020].

To summarize, the new experiments that we have added take WSIs from four distinct institutions in the Camelyon-17 dataset, extracts reasonably sized patches, and gives them a binary label based on whether the patch contains tumor tissue (as annotated by a physician). Each of the four clients (i.e. hospitals) in the FL scheme has data from a single institution, while training and test datasets have no patient overlap. We construct a joint test dataset by merging the test data from each client so that it contains mostly "external" data from the perspective of any one client. Hence, for a trained model to show good test performance it must generalize to external data.

Point 1. If I understand correctly, you make each model output predictions for 32-classes even though you only have training/test data for 26-classes? While I agree that this should not affect the relative comparison, this is confusing for readers. It probably should be simply stated as a 26-class problem in the manuscript.

Response: As the previous experiment could have been confusing, we have replaced the experiments with new ones that are now binary classification which allows us to focus on the Federated Learning aspects more clearly. We hope this point is no longer a concern. See above.

Also, is random sampling used to create train/test sets? I do not think that is the right approach in this case. Given some classes have so few cases (e.g. SARC has less than 15 cases in total), if you randomly sample your training/test splits, it is very likely some of the

minority classes will not appear at all in the test set. Can the authors clarify if this is indeed the case? i.e., I do not think it's enough that the authors just provide a count of cases in each class across the entire dataset, instead please specify how many cases are in the training and in the test of each class. This is how it is done in every major published computational pathology study and should be also followed here. After all, there is no reason to include minority classes in your training set/problem if you do not plan on evaluating on them.

Response: These are excellent points, and together with the reviewer's other suggestions it made sense to switch from the highly imbalanced TCGA dataset, to the Camelyon-17 dataset where the binary labels are more balanced. In the new experiments the train/test sets are constructed per client by assigning several patients to the test set, so that there is no patient overlap across the split, and in such a way that the number of datapoints approximates an 80/20 ratio with as much class balance as possible. The new Table 1 clearly demonstrates the splits for each client, including the count of cases in each class for all train and test sets.

Point 7. Given the large class imbalance in the histology dataset (e.g. some classes have < 15 samples in total, and some have nearly 1000), I had specifically asked for the report of macro-average (i.e. balanced accuracy) on the histology dataset. Yet for some reason the authors chose to only report balanced accuracy on the benchmark datasets (figure 12)? When the class-imbalance is this significant, the single accuracy metric is simply not sufficient to characterize the performance of the models. For instance, a model that simply predicts every case as the single predominant class will appear to do very well compared to the competition when in fact it is useless. Given that this is supposed to be an application to a real-world problem, I would suggest that the authors at least report:

- accuracy for each class
- macro-averaged accuracy over all classes

Another suggestion would be to simply remove any classes with < 10 or <5 cases in the test set from the problem, since we cannot reliably assess the model's performance on those classes.

Response: Again this is an excellent point about the previous version of the experiment which motivated us, based on your feedback, to switch to the Camelyon-17 dataset where classes are balanced, and we can focus on the FL aspects more clearly. As you can see in the new Table 1, each of the two classes is now well-represented in the train and test sets. Still, for completeness we report macro-averaged accuracy in Figure 10 along with accuracy.

Point 9. I am quite familiar with WSI-classification and both references the authors mentioned (Dual-Stream Multiple Instance Learning Network and TransMIL) do not sample patches (i.e. use a mosaic representation) the way the authors do in this work. Both approaches presumably consider all tissue regions in the slide for the classification task (similarly, the other works I mentioned, i.e. Campanella et al 2019, and Lu et al. 2021 also process/consider the entire tissue region for classification). Therefore I do appreciate that the authors added additional explanation about the method they used to the manuscript.

Response: We are glad that the previous version was clarified. In the new version we follow [Andreux et al. "Siloed Federated Learning for Multi-Centric Histopathology Datasets",

2020] which extracted patches from WSIs in the Camelyon datasets, with labels assigned to each patch from pixel-level annotations of tumors.

Point 11. As I alluded to and the authors also acknowledged, there are a number of anatomic sites in the TCGA dataset considered that only have a single associated tumor subtype and therefore the model can just infer the label from normal tissue region. It is also not a useful problem to solve since the anatomic site is presumably given. Why not just remove these classes/slides from the problem? This is probably the main reason that every computational study for classification I have seen restrict their attention to classifying between multiple subtypes that may be present at a given anatomic subtype. Since you already used multiple benchmark datasets, and want to use at least one real-world problem that is clinically relevant, then I think it would be much more appealing to readers with clinical or computational pathology background that the problem the authors proposed to solve is actually a meaningful one and is suitably framed.

Response: Thank you for the advice based on standards for computational studies. For the new Camelyon-17 experiment, all WSIs are lymph node sections from breast cancer patients. Since this is restricted to a single anatomic subtype, we hope it is in line with how previous studies have been framed.

Lastly judging from figure 9 b), the performance of ProxyFL vs. other approaches appear to be the similar under strong differential privacy guarantee, and ProxyFL actually does statistically significantly worse than CWT under weak privacy? Can authors confirm if this is indeed the case? So is the only advantage of ProxyFL faster communication then? However, looking at 9 d), it is not like ProxyFL is an order of magnitude faster either, so perhaps the authors want to summarize and make a stronger case by clarifying what are the clear advantages of the proposed method over the competing approaches.

Response: We note that the outcomes of our histopathology experiment are slightly different on the Camelyon-17 dataset, due to the changes in data and patch extraction approach in response to the preceding discussions with the reviewers. Please see Figure 10 in the new submission. We find that ProxyFL and FML have the most similar performance (consistent with the results of our other two experiments in Figs. 3 and 8), but by the end of training ProxyFL has achieved better accuracy. While all federated approaches can improve upon the non-collaborative regular baseline, we believe the improvement in accuracy and macro-accuracy from ProxyFL compared to CWT or FML is meaningful. These results apply for a relatively strong differential privacy guarantee of about $\epsilon=2.2$ (for comparison, see [Papernot et al. "Tempered Sigmoid Activations for Deep Learning with Differential Privacy" 2020] where state-of-the-art guarantees are $\epsilon=2.7$ on a dataset with more datapoints). In addition, faster communication compared to centralized approaches is certainly another advantage of ProxyFL.

In summary, previous approaches to Federated Learning for medical images are either centralized (FedAvg, FML), limiting the autonomy each institution has over its model for regulatory compliance and personalization, or give relatively weak performance when differential privacy is applied (CWT Figs. 3, 8, 10). ProxyFL natively incorporates differential privacy guarantees, is fully decentralized, and is communication efficient (Figs. 4, 10e).

Reviewer #4 (Remarks to the Author)

The authors have addressed most of the previous comments satisfactorily, with some remaining points included below.

1. With regard to Rev#2 Q#1 (terminology), the fact that the authors consider FL as a group of methods should be clearly defined in the manuscript, as otherwise might mislead the reader.

Response: Thank you for the suggestion. We followed the standard notations of centralized and decentralized FL frameworks as discussed by Li et al. (2020a) where decentralized topologies are also considered FL. We added some clarification to the introduction section.

2. In response to Rev#2 Q#2 the authors point to additional text that they provide in Section 2 "Related Work (Line 85-95)": "However, unlike our method, none of these protocols provides a guarantee of privacy for participants, and therefore cannot be used safely in highly regulated domains". It is unclear what the authors mean by "safely". Please either clarify, or reword.

Response: Thank you for the suggestion. We've reworded the sentence to avoid the word "safe" which could be misconstrued, and to emphasize our quantitative privacy guarantee which is essential for highly regulated domains.

3. In response to Rev#2 Q#3 the authors point to additional text that they provide in "Conclusion and Future Work (Line 373-380)": "Finally, while differential privacy provides theoretical guarantees on the protection of privacy based on mathematical analysis, it is important to validate the effectiveness of these guarantees in practice. To be meaningful, such guarantees should demonstrably reduce the susceptibility of systems to reconstruction and membership inference attacks [Bhowmick et al., 2018]. Recent works use realistic privacy attacks to empirically test how much privacy is afforded by DP-SGD [Jagielski et al., 2020], while Nasr et al. [2021] demonstrate worst-case attacks that achieve the maximum privacy violation allowed by the theoretical guarantees of DP-SGD, but do not breach them. Testing and validation of privacy guarantees should be incorporated into model development and deployment processes."

I would also mention here that DP guarantees in the paper use accounting based on to what degree a single data sample can affect the overall outcome. These guarantees do not speak to vulnerability of attacks that make use of correlations across the data samples. A simple example of this is the fact that $\epsilon=0$ DP ('perfect DP') will leak a lot of information about my single data sample if it is applied to a dataset that consists solely of multiple copies of that single sample (since the DP analysis only compares query outcomes with and without a single sample removed). Generally, for this reason, it is more possible than the DP guarantee may suggest extracting information from the result of a DP algorithm (as defined in the paper) when there are strong correlations regarding this information across samples. This is another way in which ϵ , δ numbers may not provide the meaning of privacy that one may expect, and the reader should be aware of this.

Response: The reviewer is correct in that when a single individual contributes multiple datapoints to a dataset, they will have a weaker privacy guarantee than expected because DP is defined in terms of adding or removing a single datapoint, not several. However, there is an established theory on how exactly the privacy guarantee degrades in this case, so-called "group differential privacy" [Dwork and Roth, The Algorithmic Foundations of

Differential Privacy, 2014]. We have added mentions of group DP to the summary of DP in the Introduction, and when we apply it in Section 4.4 to a public, real-world dataset.

Although we agree that the practical implications of DP guarantees need to be discussed carefully, and we have expanded on this in the manuscript, we respectfully disagree that the algorithm will be vulnerable to attacks/leakages in the perfect DP case mentioned by the reviewer. For instance, if we were to query the average salary of a dataset where all examples are the same person, a perfect DP algorithm will return a number from the same distribution regardless of who that person is. This is because, by definition, $\epsilon = \delta = 0$ means the distribution of outcomes must be exactly the same for **any dataset** (by induction, differing by many datapoints is the same as differing by one in the special case of $\epsilon = \delta = 0$). Nevertheless, for a non-trivial DP guarantee where ϵ is not zero, we agree with the reviewer that its practical interpretation is critical and should be tailored to the application.

4. Finally, in addition to the original review comments, I do not see the authors explaining how the adjacency matrix (P_t in Algorithm 1) is selected for the various experiments. Whether it was randomly initialized (i.e. how P_0 is chosen), constant over time (I.e. $P_t = P_0$ for all t), and/or evolved in some specified way.

Response: The adjacency matrix is based on the communication graph depicted in Fig.2. We also explained this exponential graph in more detail in the first paragraph of Section 4. As a concrete example, client #0 will send its proxy to client #1 in the first round (the neighbor that is (2^0) -steps away), and in the second round, client #0 will send its proxy to client #2 (the neighbor that is (2^1) -steps away). Every client will do a similar communication as shown in Fig.2 so that information can be quickly propagated to the whole population. Hence, the adjacency matrix we use is evolved in a specific periodic way, and not randomly initialized or constant over time.

REVIEWERS' COMMENTS

Reviewer #1 (Remarks to the Author):

Dear authors, thank you for addressing my concerns on the use of TCGA pan-cancer diagnosis as a benchmark and switching to C17. My concerns have been addressed. As a final note, a few places in the manuscript (e.g. abstract/section 4) still speak of the pan-cancer diagnosis task but I trust that it will be corrected in the final version.